# Assessing Chronodisruption Distress in Goldfish: The Importance of Multimodal Approaches

**DOI:** 10.3390/ani13152481

**Published:** 2023-08-01

**Authors:** Nuria Saiz, Lisbeth Herrera-Castillo, Nuria de Pedro, María Jesús Delgado, Sven David Arvidsson, Miguel Ángel Marugal-López, Esther Isorna

**Affiliations:** Department of Genetics, Physiology and Microbiology, Faculty of Biological Sciences, Complutense University of Madrid, 28040 Madrid, Spain; nursaiz@ucm.es (N.S.); lisbethh@ucm.es (L.H.-C.); ndepedro@ucm.es (N.d.P.); mjdelgad@ucm.es (M.J.D.); svenarvidsson@outlook.es (S.D.A.); mmarugal@ucm.es (M.Á.M.-L.)

**Keywords:** circadian system, anxiety, behavioral test, fish, stress, cortisol, welfare, open field, thigmotaxis, scototaxis

## Abstract

**Simple Summary:**

The disruption of circadian rhythms is considered a potential source of distress, but the extent of its consequences is under study. This research investigated whether the disruption of the light/dark and feeding/fasting cycles can trigger stress and anxiety-like behavior in fish. For this purpose, we first optimized behavioral tests to measure anxiety-like states in goldfish. Second, we studied anxiety and stress responses in two groups of goldfish exposed to either continuous light or randomly scheduled meals for two months. Both conditions led to anxiety-like behavior, evidenced by increased thigmotaxis in the open field with object approach task. Fish exposed to constant light also showed higher locomotor activity, suggesting greater energy expenditure. Additionally, chronodisruption led to increased cortisol levels throughout the experiment, with evidence of a possible axis fatigue due to chronic stress in fish under continuous light. Altogether, these findings support that these chronodisruptive conditions cause distress in the animals, and they should be considered for welfare improvement. The correlation analyses suggested that cortisol, thigmotaxis, and scototaxis are not dependent on each other, while there is a high influence of individual behavioral traits. Thus, the need for combining multiple parameters when assessing discomfort in fish is emphasized.

**Abstract:**

Chronodisruption caused by factors such as light at night and mistimed meals has been linked to numerous physiological alterations in vertebrates and may be an anxiogenic factor affecting welfare. This study aims to investigate whether chronodisruption causes measurable changes in the anxiety responses of goldfish under two conditions: randomly scheduled feeding (RF) and continuous light (LL). Anxiety-like behavior was assessed in the open field with object approach and black/white preference tests, which had been validated using diazepam. An increased thigmotaxis response and decreased object exploration under both chronodisruption protocols indicated anxiety states. Furthermore, locomotor activity was increased in LL fish. The black/white preference test discriminated anxiolysis induced by diazepam but was unable to detect anxiety caused by chronodisruption. Plasma cortisol increased in both RF and LL fish throughout the experiment, confirming that both conditions caused stress. The LL fish also showed an apparently desensitized hypothalamus–pituitary–interrenal HPI axis, with a decrease in *pomc* and *crf* expression. Individual analysis found no correlation between anxiety-like behavior and stress axis activation nor between scototaxis and thigmotaxis responses. However, individual differences in sensitivity to each test were detected. Altogether, these results highlight circadian disruption as a stressor for fish and endorse a multiple variable approach for reliably assessing animal discomfort.

## 1. Introduction

Due to the Earth’s rotation, living organisms are exposed to substantial fluctuations in their environment that follow a 24 h cycle. To cope with these predictable changes, animals have evolved an endogenous timekeeping mechanism known as the circadian system that allows them to anticipate and adapt their physiology and behavior [1,2]. This system consists of an intricate network of cell-autonomous oscillators found in most tissues sustained by transcriptional–translational feedback loops of clock genes with a period of approximately 24 h [1,2]. These loops regulate the expression of a plethora of genes involved in various physiological functions leading to neural, hormonal, and metabolic circadian rhythms that can persist for several days in constant conditions. The optimal coordination of these biological rhythms is the basis for temporal homeostasis [3,4]. 

Circadian oscillators adjust their phase and period to external inputs (also known as Zeitgebers) such as variations in light, temperature, or food availability in a process called entrainment [4]. Light is generally considered as the main Zeitgeber because in mammals it entrains clock gene expression in the suprachiasmatic nucleus, a master oscillator that synchronizes the rest of the oscillators [5]. In fish, however, the existence of a master pacemaker has not been demonstrated, and their system is best described as a network of interconnected oscillators with varying sensitivities to each synchronizer [6,7]. Nonetheless, light is likely the main circadian input for most teleosts as well [8], as neural pacemakers in fish are predominantly driven by the photocycle [9,10]. Food availability is also a key Zeitgeber, especially for peripheral oscillators, which are entrained by regular feeding time in both fish and mammals, regardless of the photocycle [9,10,11,12,13]. Depending on their reliance on each cue, several oscillators in fish have been characterized as either light-entrainable (LEOs) or food-entrainable oscillators (FEOs) [6,7,14]. However, the synchronizing factors that entrain the circadian system can become desynchronizers when they are untimely. When this occurs, the temporal organization of physiological, metabolic, and behavioral processes is disrupted, leading to stress, adverse health effects, and increased mortality, as is well documented in mammals [15,16]. In these vertebrates, artificial light at night (ALAN) blunts the rhythms of clock genes, glucocorticoids, and locomotor activity, causing alterations in metabolic function, cognitive disorders, and cancer [17]. Similarly, mistimed meals, such as night-time eating, have been associated with the onset of metabolic syndrome [18]. In fish, ALAN has been found to affect growth, cause alterations in the immune system, retina degeneration, and tumor formation, among others [19,20,21]. In fact, ALAN is an emerging challenge for the conservation of aquatic ecosystems [22]. Other models of desynchronization, such as uncoupling daily light–dark cycles and feeding time, caused disrupted clock functioning as well as an increase in and/or loss of cortisol rhythms [13], suggesting a conserved crosstalk between circadian system and stress in fish. The uncoupling of Zeitgebers has been associated with anxiety in mammals [23], but it is not yet clear whether chronodisruption also affects fish welfare as an anxiogenic factor or if the stress axis activation and anxiety-like behavior are correlated.

The physiological stress response involves the activation of the hypothalamus–pituitary–adrenal neuroendocrine axis, leading to the production and release of glucocorticoid steroid hormones by the adrenal tissue, which trigger metabolic and neural pathways that increase energy mobilization and alertness, thereby improving the response to environmental hazards [24,25,26,27]. As in mammals, chronic stressors cause a sustained elevation of cortisol levels, making this hormone a reliable indicator of fish welfare [24,28,29]. Moreover, beyond being the major stress hormones, glucocorticoids have a role in maintaining temporal homeostasis. Under unstressed conditions, daily glucocorticoid secretion is rhythmic in most species studied, including several teleosts, with higher levels during the wake phase and a peak just before the onset of activity [21,28]. For this reason, glucocorticoids are considered one of the main endocrine outputs of the circadian system [7,30]. This rhythm of glucocorticoids is parallel to that of pituitary adrenocorticotropic hormone (ACTH), which is stimulated by the corticotropin-releasing factor (CRF) released by the hypothalamus [31]. Notably, glucocorticoids can entrain and phase-shift clock gene expression in the peripheral tissues of fish and mammals [32,33]. Because of this dual role as outputs and input of the circadian system, they are suggested to be internal synchronizers that participate in the oscillators’ alignment [34,35].

Anxiety is an adaptive neural and physiological response triggered by a perceived threat in the environment. Environmental stressors, such as exposure to novelty, isolation, movement restriction, predators, or certain chemicals, can induce anxiety-like behaviors in fish that are comparable to those observed in mammals [36,37]. One of these behaviors is scototaxis, the preference for dark over light environments, which is present in many species, including teleosts [38,39,40]. The black/white preference test has been developed as a tool to assess the tendency of fish to prefer the dark half of a black and white narrow arena, which is associated with anxiety levels. This way, anxiolytics like benzodiazepines or ethanol decrease this preference in adult zebrafish [41,42]. Another behavioral test employed in fish is the open field test with object approach, where an unfamiliar object is placed in the center of a proportionally ample tank. In this test, thigmotaxis opposed to tendency to exploration (i.e., “boldness”) is assessed. Thigmotaxis (or “wall-hugging”) is the tendency of animals to stay close to the wall, which is an indicator of anxiety in both fish and mammals and is also offset by anxiolytic drugs in the zebrafish [43,44]. The assessment of these anxiety-like behaviors is useful for understanding the effects of stressors on fish welfare and identifying potential interventions to mitigate their negative impact.

The present work aims, firstly, to evaluate the reliability of two behavioral tests, the black/white preference and the open field with object approach tasks, as indicators of anxiety in the goldfish, *Carassius auratus*. Secondly, we explored the anxiety-like responses of goldfish to two chronodisruption models, namely the absence of a light–dark cycle (24 h constant light) and the absence of a feeding–fasting cycle (irregular feeding schedule). Finally, we have studied whether behavioral changes are associated with alterations in plasma glucocorticoids, the possible relationship between the thigmotaxis and scototaxis responses, and whether there is a consistency in the behavior of individual animals when they repeat the tests.

## 2. Materials and Methods

### 2.1. Animals and Housing

Goldfish (Carassius auratus) were acquired from the commercial house ICA S.A. (Madrid, Spain) and housed in 60 L fish tanks (8–10 individuals/tank), with filtered and aerated water at 21 ± 1 °C. Unless otherwise indicated, fish were kept under a controlled photoperiod (12 h of light and 12 h of darkness, 12L:12D) and fed once daily at the beginning of the photophase (ZT 1 or ZT 2 h, depending of the experiment; ZT 0 h being the start of the photophase) by automatic feeders with commercial granulated feed (1.5% body weight, bw, Sera Pond Biogranulat, Heisenberg, Germany). All fish were tagged with subcutaneous injections of black ink (Eternal Ink, Brighton, MI, USA) for individual identification. Animals were acclimated to these conditions for 20 days prior to the experiment. All experiments complied with the Guidelines of the European Union Council (UE63/2010) and the Spanish Government (RD53/2013) for the use of animals in scientific proposals and were approved by the Animal Experimentation Committee of the Complutense University and the Community of Madrid (PROEX 107/20).

### 2.2. Experimental Designs

#### 2.2.1. Protocol Optimization: Effects of Diazepam on Anxiety-Like Behavior

To determine the reliability of the behavioral tests for measuring anxiety-like behavior in goldfish, fish were treated with diazepam (Normon, Madrid, Spain), an anxyolitic drug, and various variables were measured (see Section 2.3). Each day before the experiment, diazepam (5 mg) was dissolved in 1 mL of vehicle (composed of 75% distilled water, 25% DMSO, Sigma-Aldrich, St. Louis, MO, USA). This stock solution was diluted at 1:10 (5 μg/g bw dose) or 1:5 (10 μg/g bw) for injection. Goldfish (5.49 ± 1 g bw) were divided into 3 experimental groups and injected intraperitoneally (IP; 10 μL/g bw) at ZT 2 h with the vehicle (2.5% or 5% DMSO in distilled water, control groups, *n* = 16, respectively), diazepam 5 (5 μg/g bw, *n* = 8), and diazepam 10 (10 μg/g bw, *n* = 8). Doses were selected based on previous works in the same species [45]. All injections were performed under deep anesthesia (tricaine methane sulfonate, MS-222, 0.14 g/L, Sigma-Aldrich). After injection, fish were isolated for 20 min in a 5 L tank in the room where the test was to be performed to allow them to acclimate and the drug to take effect. After that, the open field with object exploration and the light/dark preference tests were performed, as indicated in Section 2.3. 

#### 2.2.2. Effect of Chronodisruption on Stress and Anxiety-Like Behavior

Goldfish (4.4 ± 0.1 g bw) were divided into 3 experimental groups (*n* = 16–20, 2 tanks/experimental group). All groups were fed once per day with automatic feeders (1.5% bw) under different photoperiodic conditions and feeding times (Figure 1): (1) control group: maintained under acclimation conditions (12L:12D and daily fed at ZT 1 h); (2) random feeding (RF) group, maintained under 12L:12D photoperiod and fed once daily at a random time (generated with the RAND function of Microsoft Excel 2016^®^, Microsoft, Redmond, WA, USA); and (3) continuous light (LL) group, maintained under 24 h of light (24 L) and daily fed at the same time as the control group, i.e., circadian time CT 1 h (as this group was under free running conditions, a subjective time scale should be considered, CT 0 being the time of the last light onset). Fish were weighed once a week to readjust the amount of food that needed to be dispensed and ensure that 1.5% was being provided.

Cortisol and behavior were measured after short-term exposure to chronodisruption (days 3–4). Behavior was measured again after long-term exposure (days 46–47), along with stress-related parameters (cortisol and *pomc* (ACTH precursor) and *crf* mRNA abundance, day 53), as indicated in Figure 1.

All the tests were performed sequentially during the ZT 6 and ZT 10 (or CT 6 and CT 10 for LL group) time interval, testing a maximum of 8 animals per day, always on animals fed at ZT 1 (or CT 1) to avoid a heterogeneous fasting period. On days 3–4, immediately after the behavioral tests, blood was obtained from the caudal vein of anesthetized animals (MS-222, 0.14 g/L) with heparinized syringes and centrifuged at 6000 rpm for 5 min. Plasma was stored at −20 °C until cortisol analysis. On the last day of the experiment (day 53), blood was collected at ZT 5 h, and thereafter, fish were sacrificed by anesthetic overdose (MS-222, 0.28 g/L) and the hypothalamus and pituitary were sampled, frozen in liquid nitrogen, and stored at −80 °C until they were processed for analysis of mRNA abundance (*crf* in hypothalamus and *pomc* in pituitary).

### 2.3. Behavioral Tests

#### 2.3.1. Open Field and Object Approach Test

The open field/object approach test (OF) was employed to evaluate anxiety-associated behaviors, such as thigmotaxis and non-exploration of novel objects. A custom-made circular tank of transparent methacrylate and 50 cm diameter, filled with water to a depth of 10 cm, was used. The bottom of the tank was white to allow appropriate contrast for the tracking, and the wall was covered with opaque paper. The novel object consisted of a yellow and blue toy of 7 cm in height and 4 cm in diameter placed in the center of the arena (Appendix A). The tank was illuminated with white light of 200 lux intensity on the surface of the water, and the temperature was the same as in the housing tank (21 ± 1 °C). The open field area was considered the inner part (occupying 75% of the total area) while the outer area (25%) was considered “close to the wall” (indicative of thigmotaxis, Appendix A). In addition, the central area surrounding the object, occupying 25% of the total area, was considered the novel object zone (Appendix A). Fish were moved individually from their tanks to the testing room in 5 L plastic tanks in which they were acclimated for 5 min. Then, they were placed near to the tank’s wall, and their trajectory was recorded with a video camera for 10 (2.2.2) or 20 (2.2.1) minutes. The water was renewed between fish coming from different aquaria. The tests were recorded using video cameras, and behavioral indicators were acquired from automated tracking of the videos using Ethovision XT 17 software (Noldus, Wageningen, The Netherlands). The analyzed parameters were latency to first entrance to open field, number of entries to the open field, time spent in open field, latency to first object approach, and average swimming velocity.

#### 2.3.2. Black/White Preference Test

The black/white preference task (BW) test was employed to assess scototaxis. A rectangular container made of non-reflecting methacrylate which was 47 cm long, 14.5 cm high, and 10 cm wide was used, half of which (23.5 cm long) was black and the other half white (Noldus; Appendix A). The testing tank was filled with filtered water to a depth of 10 cm (4.5 L), and the light intensity was adjusted to 600 lux at the water surface. The temperature was maintained as in the housing tank (21 ± 1 °C). The water was exchanged between individuals from different tanks. After undergoing the open field test, individual fish were returned to the 5 L transporting tanks and acclimatized to the black/white testing room for 5 min. Fish were then placed inside a separator in the center of the black–white container (Appendix A) and allowed to acclimate for 3 min. Once this time had elapsed, the separator was lifted, allowing the fish to swim freely for 15 min while they were recorded with a video camera. Behavioral data were acquired from automated tracking of the videos using Ethovision XT 17 software (Noldus). The analyzed parameters were latency to white, number of crossings between black and white zones, time spent in the white zone, and average swimming velocity. 

### 2.4. Plasma Cortisol Levels

Plasma cortisol was measured in duplicate as previously reported [46]. Briefly, 3N HCl was added 1:1 to samples and allowed to stand for 10 min at room temperature. After that, methanol was added (1:10, volume sample: volume methanol) and the tube was centrifuged (12,000 rpm) for 10 min twice, collecting the supernatants in which cortisol remains. Cortisol was quantified in the supernatants using an ELISA assay (Cortisol ELISA Kit, Cayman Chemical, Ann Arbor, MI, USA) according to the manufacturer’s instructions. Free cortisol values were within the range described by the manufacturer (10–800 ng/mL). 

### 2.5. Gene Expression Analysis

The total RNA (from hypothalamus and pituitary) was obtained using TRI^®^ Reagent (Sigma-Aldrich, St. Louis, MO, USA) following the protocol provided by the manufacturer. RNA was treated with RQ1 RNase-Free DNase (Promega, Madison, WI, USA) to remove genomic DNA. cDNA was obtained using SuperScript IV Reverse Transcriptase (Invitrogen, Waltham, MA, USA), 0.3 µg total RNA, random primers (Invitrogen, Waltham, MA, USA), and RNase inhibitor (Promega, Madison, WI, USA). The RT-qPCR was carried in duplicate in a CFX96TM Real-Time System (Bio-Rad Laboratories, Hercules, CA, USA), using iTaqTM Universal SYBR Green Supermix (Bio-Rad Laboratories, Hercules, CA, USA). PCRs were run on 96-well plates loaded with 1 µL of cDNA and 0.5 µL of forward and reverse primers (Sigma-Aldrich, St. Louis, MO, USA) 10 µM (Appendix A), to a final volume of 10 µL, as previously described [47]. Each PCR plate also included a standard dilutions curve to ensure the efficiency of PCR reactions (90–105%) and water and pre-RT RNA as negative controls. The RT-qPCR protocol consisted of 1 cycle at 95 °C for 30 s, followed by 40 cycles of a two-step amplification program (95 °C for 5 s and 60 °C for 30 s). A melting curve was generated (temperature gradient at 0.5 °C/5 s from 70 to 90 °C) to verify the specificity of amplified targets. To determine relative mRNA expression (fold change), we used the 2^−∆∆Ct^ method [48], considering that the mean of the control group has a relative value of “1”.

### 2.6. Statistical Analysis

Behavioral indicators of anxiety and relative gene expression data are represented in vertical bar graphs as mean + standard error of the mean (SEM). Data series of diazepam-treated fish (5 and 10 μg/g bw) were compared with ANOVA (followed by the Student–Newman–Keuls post hoc test). Chronodisrupted (LL and RF) groups were compared with their respective controls using a Student’s *t*-test. Normality and homoscedasticity of data were confirmed using the Shapiro–Wilk and Levene tests, and data were adjusted to a logarithmic or square root scale when necessary. When data did not meet these requirements, the non-parametric Mann–Whitney U test was used. Correlation analysis between variables (behavioral parameters vs. cortisol, behavioral parameters in black/white vs. open field, and anxiety indicators on day 3–4 vs. day 45–46) was performed using the Spearman rank-order test. The significance threshold *p* = 0.05 was considered for all tests. The correlation coefficients were also calculated, with CC > 0 meaning that the variables tend to increase together and CC < 0 meaning that one variable decreases when the other increases. All the statistical tests and transformations stated, as well as the graphical representation of the data, were performed using SigmaPlot 12 software. 

## 3. Results

### 3.1. Effects of Diazepam on the Open Field and Black/White Tests

Control goldfish showed thigmotaxis in the OF test, spending about 80% of the time near the walls (25% outer area) and only 20% of the time in the open field (75% remaining inner area) (Figure 2a and Figure 3c). This thigmotaxis was reduced by the administration of diazepam, with fish spending twice the time in the open field area after both drug doses (5 µg/g bw and 10 µg/g bw) (Figure 2a–c and Figure 3c). The latency in entering the open field zone tended to be lower (but not significant statistically) in diazepam-treated animals (Figure 3a), and fish treated with the low diazepam dose tended to enter this zone more frequently (Figure 3b). Diazepam treatment (the higher dose, 10 μg/g bw) also reduced the latency to enter the central zone to explore the object, Figure 3d. In addition to decreasing anxiety-like behavior, diazepam at the higher dose tended to affect the locomotor activity of fish, causing them to move at a lower average speed (Figure 3e) and to move significantly less frequently between zones (Figure 3b).

In the black/white preference test, the control fish spent less than 30% of the time in the white half of the tank (Figure 2d and Figure 4c). This scototaxis is reduced by diazepam administration, which increased the time spent in the white zone with the 10 µg/g bw dose (Figure 2d–f and Figure 4c) and tended (no significant differences) to decrease the latency to enter the white zone (Figure 4a). Finally, as in the open field test, a tendency to move slower was observed under 10 µg/g diazepam (Figure 3d), while the number of crosses between zones was not modified (Figure 3b).

### 3.2. Chronodisruption Effects on the Open Field Test

In the OF test, the fish under continuous light (LL) entered the open field fewer times than the controls on both days 3–4 and 45–46 of the experiment (Figure 5c,d) and spent less time in this area on day 45–46 (Figure 5f and Appendix A), suggesting an enhanced thigmotaxis. LL fish also showed a higher latency for the first novel object approach (Figure 5g). Despite the reduced exploration, fish under LL moved at a higher velocity during the test than the controls after 3–4 days and 45–46 days (Figure 5i,j). On the other hand, the randomly fed fish took significantly longer to leave the wall zone for the first time on days 3–4 and 45–46 (Figure 5a,b). These RF fish also entered the open field zone less frequently (Figure 5c,d; significantly on days 3–4) and spent less time there than the controls (Figure 5e,f; significantly on days 45–46). However, the RF fish showed no significant differences regarding the latency to central area (object approach).

### 3.3. Effects of Chronodisruption on Black/White Test

In the BW test, fish under constant light showed a tendency (non-significant) to change zone less frequently at both 3–4 and 45–46 days (Figure 6c,d), but the latency and the time in the white zone were similar to those of the controls (Figure 6a,b,e,f). None of the behavioral parameters were significantly modified by 3–4 days of randomly scheduled feeding (Figure 6a,c,e,g). However, when the test was performed after a longer time of RF conditions (45–46 days), fish took slightly less time to enter the white zone, crossing more frequently between zones and spending more time in the white zone than the controls (Figure 6b,d,f). Velocity was similar among groups or times (Figure 6g,h). Fish move at a lower velocity in the BW test than in the OF test (Figure 5i,j).

### 3.4. Effect of Chronodisruption on the Neuroendocrine Stress Axis

Regarding the HPI axis, circulating cortisol was not modified during the first 3–4 days of chronodisruptive conditions, with similar values in fish from the three experimental groups (Appendix A). At the end of the experiment (day 53), both chronodisruptive conditions (RF and LL) had caused an increase in plasma cortisol, while the controls maintained values near to the initial ones (Figure 7a). In addition, a significant decrease in the expression of pituitary *pomc* and hypothalamic *crf* was observed in the LL group (Figure 7b,c).

### 3.5. Correlations among Behavioral Parameters at Different Times in the Same Individual, among Parameters of Both Tests, and among Behavioral Indicators and Cortisol Levels

The Spearman tests showed that plasma cortisol levels were not correlated significantly with any of the behavioral variables studied in the OF and BW tests when analyzed by individual (Table 1 and Table 2). There was also no correlation between most thigmotaxis-indicating parameters from the OF test and scototaxis-indicating parameters from the BW test (Appendix A). Interestingly, the average velocity in the OF test is correlated to the number of crossings in the BW test, as well as the velocity of the same fish in both tests (Appendix A). The behavior of each individual on days 3–4 and on days 45–46 was significantly correlated (Table 3 and Table 4). This way, we found that there is a correlation in the number of entries to the open field, the latency to the object, and the average velocity in the two performances of the OF test by the same fish (Table 3). A correlation was detected in the latency to white and the average velocity (near significant) of specific individuals (Table 4) from the first to the second BW test. 

## 4. Discussion

We have examined the effects of chronodisruption on stress and anxiety in goldfish. Despite cortisol being the classical welfare indicator in fish [28,29], the use of behavioral tests to assess the effects of diverse factors on anxiety-like behavior is rapidly increasing in teleosts [37,49]. These tests offer several advantages, such as being non-invasive, quick, and affordable [49]. In this work, we have optimized two behavioral tests to evaluate scototaxis (black/white test) and thigmotaxis (open field with object approach test), supporting their relevance in the analysis of emotional responses in goldfish. Moreover, our results demonstrate that HPI activation and anxiety are independently enhanced in two models of chronodisruption—the absence of light/dark and feeding/fasting cycles—in this teleost.

The evaluation of scototaxis (in the black/white preference test) and thigmotaxis (in the open field test) shows the result of a conflict between the safeness that a dark area or a wall provides to the animals and their natural motivation to explore new environments [37,50]. Several fish species are known to exhibit scototaxis and thigmotaxis, including zebrafish, common carp, salmon, three-spine stickleback, Mexican blind cavefish, medaka, and goldfish [50,51,52,53,54,55,56,57], although these behaviors can vary during the earlier developmental stages [57,58]. In this work, untreated goldfish spent around 80% of their time near the walls in the OF test and 70% of their time in the black zone in the BW test, confirming the presence of these behavioral tendencies, as expected in adult goldfish. However, since the specific characteristics of each testing method can affect the results [59], the anxiolytic drug diazepam [60] was used to confirm the reliability of the employed behavioral parameters as anxiety indicators. In the OF test, diazepam-injected fish spent significantly more time at the center and approached the object earlier than the controls, indicating a drug-induced reduction in thigmotaxis and increased exploration, as expected. This effect of diazepam to reduce time spent near the wall in the open field has been established in mammals [61] and zebrafish [43] and reproduced in goldfish with Tofisopam, another anxiolytic drug [52]. The average swimming velocity and the number of entries to the open field were not affected by the lower dose of diazepam, but the higher dose decreased both parameters, which was probably due to the sedation and myorelaxation effects of diazepam that affect general locomotor activity [60]. Regarding the BW test, previous studies in goldfish have shown that diazepam reduces the time it takes for fish to enter the white zone for the first time [62]. The diazepam-treated goldfish had a slightly shorter latency to white than the controls. However, this effect was not statistically significant, likely because the fish started the test in the middle of the tank (and not in the black chamber); thus, some entries into the white zone were coincidental. In any case, the highest dose of diazepam drastically decreased scototaxis, increasing the time spent on the white side of the tank to the point that fish spent around 50% of the time on each side, in agreement with previous results in zebrafish and goldfish [54,63]. Unlike in the OF test, the high diazepam dose (10 μg/g) did not significantly affect general locomotor activity in the BW test (measured as the average swimming speed or the number of crossings between zones), although it tended to decrease. This difference is probably due to the smaller size of the BW arena. Altogether, the present results corroborated the validity of the methods used for measuring anxiety-like behavior in goldfish, as anxiety-related parameters were reduced by the diazepam treatment. Time spent in the defined zones of the OF and BW tests as well as the latency to approach the object in the OF test are the variables that show the most evidence of the anxiolytic effect.

Once the tests for measuring anxiety-like states had been implemented, the next goal was to assess the anxiety levels of fish under chronodisruptive conditions. These two models for chronodisruption have been previously well studied in the same species, causing alterations in locomotor activity, plasma cortisol, and clock gene expression rhythms. These alterations are particularly prominent in the central nervous system, especially under LL conditions, but are also observed in the periphery in randomly fed animals [9,13,64,65,66,67]. Goldfish exposed to continuous light showed increased anxiety-like behavior. After 3–4 days of constant light, fish entered the open zone of the OF test less frequently and tended to spend less time in this area, indicating an increased thigmotaxis. The latency to approach the object was also increased in this group of fish, suggesting a reduced willingness to explore (neophobia), which is an anxiety indicator [44]. On days 45–46, a similar increase in thigmotaxis was observed, with fewer entries to the open field and less time spent in this zone. Surprisingly, the latency to approach the object was equivalent to the controls. The present results differ from those found in coho salmon, where behavior on the OF test was not modified by LL conditions [64]. Species-specific factors may play a role in these discrepancies, as well as the duration of the conditions, which was shorter in our study. In rodents, depending on the report, chronic continuous light has been shown to induce anxiety-like behaviors [65,66] or to reduce them, presumably due to stress adaptation [67,68]. Another observed effect of continuous light in goldfish was the increase in the fish mean swimming velocity in both performances of the OF task (i.e., days 3–4 and 46–47). Similarly, extended photoperiods have been shown to induce hyperactivity and greater distances moved in the OF and in the plus maze tests in rodent models [69,70] and to increase general locomotor activity in mice and fish [71,72,73,74]. However, some reports on rats have obtained different results in which LL did not increase ambulation in the OF test [65].

It seems that a longer photophase can generate hypermobility, but the underlying mechanisms remain unclear, although higher levels of plasma glucocorticoids have been proposed [71]. Here, plasma cortisol was increased in fish under continuous light, but we found no correlation between swimming velocity in the test and plasma cortisol levels at the individual level. In contrast to the heightened thigmotaxis that was observed in the OF test, the BW test showed no significant effect of LL conditions on scototaxis in goldfish, in disagreement with reports in zebrafish that have shown increased scototaxis under continuous light conditions [72]. A possible explanation could be a desensitization to the aversive effect of light after being acclimated to a constantly illuminated environment. Alternatively, the enhanced locomotor activity of goldfish could cause them to move indiscriminately between sides of the arena, overriding the white avoidance, as reported in rats with a shifted light/dark cycle [75].

The link between anxiety-like behavior and light chronodisruption discussed above has been somewhat explored in mammals and a few fish species, but there is a lack of knowledge about the effects of mistimed meals on anxiety-like behavior in vertebrates since these protocols often focus on the consequences on metabolism [76]. Here, we show that fish chronically fed on a random schedule showed a heightened anxiety-like behavior in the OF test. Randomly fed fish had a higher latency to the open field zone than the controls and spent less time in this area during both studied times. Based on this, we expected that randomly fed fish would also show increased scototaxis in the BW test, but the opposite was observed. After 45–46 days, fish in the RF group spent more time on the white side and entered this zone more often than the controls. It is unlikely that the apparently reduced scototaxis in RF fish is due to lower anxiety levels. This is contradicted by the results in the OF test and the increased cortisol in these animals, which is supported by previous reports in the same species with mistimed meals. [9,13]. A possible hypothesis could be that RF fish develop a heightened willingness to explore (boldness) because they cannot predict when food will be provided. They may be in a continuous state of alertness and food anticipation [77], causing them to pass through the white and black chambers more indiscriminately in their constant food searching. This is not the case for fish fed on a regular schedule in which this search is limited to a few hours before the expected meal, what is known as food anticipatory activity [78]. Overall, the BW test does not reflect a greater anxiety-like behavior in either of the chronodisrupted groups.

The anxiogenic effect of constant light and meals given at random times could be due to a disruption of biological clock function, or the conditions themselves could be considered a chronic stressor. The fact that the two types of chronodisruption assayed (RF and LL) altered different behavioral and physiological parameters suggests that the mechanisms affected were not exactly the same. This is not surprising, given that the circadian oscillators that become dysfunctional in the absence of a light/dark or a feeding/fasting cycle are different. The central clocks depend highly on photoperiod (LEOs), while the peripheral clocks are more sensitive to mealtime (FEOs) [9,79,80]. Hence, the previously proven negative effects of the absence of photoperiod and of a fixed feeding time on the circadian system of goldfish [9] seem to also extend to behavior.

We also studied the activation of the HPI axis, one of the main indicators of welfare in fish [29]. Chronic disruption of the circadian system by removing one of the two main Zeitgebers (photoperiod and feeding schedule) caused an increase in plasma cortisol levels throughout the course of this experiment. This suggests that an unpredictable feeding time and the absence of a daily light/dark cycle are stressful conditions for goldfish, which is supported by previous results [13]. The relationship between chronodisruption and the activation of the stress axis is complex. In fish, lengthening the photophase increases plasma cortisol levels in pancake batfish and African sharptooth catfish [81,82]. Random feeding also increases plasma glucocorticoids in gilthead seabream and goldfish [9,83]. However, different results were reported in the Senegalese sole, which showed lower mean values under LL due to the disappearance of the cortisol peak [21]. Similarly, the effects of continuous light on the stress axis in rodent models have been found to be diverse. Depending on the study, ALAN can blunt glucocorticoid rhythm or not, and both decreases and increases in plasma levels are described [17,66,84]. This suggests that ALAN has undesirable effects on glucocorticoid regulation, but the specific effect may vary depending on the model and conditions. A possible explanation for these conflicting results could be the different adaptation levels of the stress axis. Stress is generally associated with higher cortisol levels, but sometimes, prolonged stress may result in decreased levels of this hormone or its precursors as a known response of desensitization and dysregulation of the axis takes place, named adrenal fatigue [85]. In agreement, fish kept under constant light showed a reduction in hypothalamic *crf* and pituitary *pomc* expression, which may be due to negative feedback on the pituitary and hypothalamus by the increased circulating cortisol. However, this downregulation was not observed in randomly fed fish, even though they showed increased cortisol levels during the photophase. This could be explained by a possible lack of the typical cortisol decline during the night in the LL fish, resulting in a stronger desensitization of the HPI axis. Such misalignments between circulating ACTH and cortisol, associated with changes in adrenal sensitivity to ACTH, are common consequences of chronic stress [86], supporting again that the chronodisruption protocol employed is stressful for goldfish.

Following the observation that LL and RF conditions increase both thigmotaxis and cortisol levels, we analyzed the individual data from each fish to explore possible links among the parameters from the behavioral tests and plasma cortisol. No correlation was found, indicating that all behavioral parameters, including mobility, were not proportional to plasma cortisol. Aligning with this, several reports imply a disconnection among anxiety-like behavior and glucocorticoid levels in mice and zebrafish [67,87]. Together, these findings suggest that the elevation in glucocorticoids is not directly related to the level of anxiety of the animals. The anxiety-like response probably results from a complex neural integration, where glucocorticoids may only play a minor role.

Furthermore, most parameters from both behavioral tests (BW and OF) performed by the same fish on the same day do not correlate with each other, aligning with reports in rats [88] with only two exceptions. The average velocity was correlated in both tests, which cannot be attributed to either thigmotaxis or scototaxis; rather, it reflects the individual mobility of each fish. The fact that this average velocity also correlates with the zone crossing in the BW test indicates that more active fish also tend to change zones more often. However, the parameters directly linked to scototaxis or thigmotaxis were not correlated. We have found no account of a comparison between these two behavioral tendencies in other fish species, and the scarce studies that tested the correlation between scototaxis and geotaxis in zebrafish have reached conflicting conclusions [89,90]. Our results suggest that both BW and OF tests are effective indicators of anxiety after averaging an adequate number of animals, but there is a dissociation among both responses, meaning that individuals that display a higher scototaxis do not necessarily display a higher thigmotaxis under the same conditions. Then, the triangulation of approaches (i.e., the use of multiple tests to measure the same construct) can be very useful in increasing the robustness of the description of anxiety-like states, as previously suggested [90].

The hypothesis that fish have individual tendencies to show greater or lesser aversion to open spaces, novel objects, or light is reinforced by the fact that multiple behavioral parameters are replicated when the same fish performs the test twice, approximately 40 days apart. For instance, there is a correlation between the number of entries to the open field, the latency to object approach, and the average swimming velocity measured in the same fish in the OF test on days 3–4 and 45–46 of conditions. The same is observed in the case of the latency to leave black in the BW test. Thus, regardless of whether they have undergone chronodisruptive conditions, and independently of the effects that these conditions may have had on behavior, the fish here show behavioral tendencies that are characteristic of the individual. Aligning with this, some teleost models exhibit individual variation in behavior, leading to the classification of fish populations as proactive or reactive [91], with proactive fish showing higher “boldness”, i.e., willingness to explore. However, the division of fish into these two categories cannot be applied to the present scenario as the same fish can show proactivity in one test but not in the other. The individual neurophenotype of fish is likely determined by a complex combination of several genes and unpredictable environmental effects. Furthermore, even clonal fish reared in near-identical conditions show behavioral individuality in the open field test [92]. The fact that the present study did not pre-select individuals based on their character (what would have reduced variability) means that the reported behavioral effects of chronodisruption are the most overt and general, while more subtle effects may have gone undetected.

Overall, the findings of this study show that the absence of either the light/dark or the fasting/feeding cycles increases stress levels and thigmotaxis (anxiety-like behavior) in goldfish. The effects of continuous light are more pronounced than those of unpredictable meals, also producing an increase in mobility that could translate into higher energy expenditure. Together with the literature, the results seem to suggest that the two main indicators of distress in fish, the increase in cortisol and in anxiety-like behavior, are not mutually dependent, so the measurement of both is desirable for judging the status of the animals. Similarly, a battery of behavioral tests to assess anxiety in fish is preferable, as the level of challenge of each task seems to be highly dependent on the individual. In general, the fact that besides disrupting gene clock oscillation, circadian disruption is a stressor for fish, highlights the importance of considering the problem of mistimed environmental cues in the quest for greater welfare.

## 5. Conclusions

Chronodisruptive conditions, such as the absence of a light–dark cycle or a random meal schedule, increase anxiety-associated behaviors in the open field with object approach test, accompanied by an increased endocrine response to stress. However, these behavioral parameters of fish are not directly related to cortisol levels.

Both the open field with object approach and the black/white preference tests are useful tools to analyze anxiety-like behavior. Different fish show different susceptibilities to these tests, revealing that the responses of thigmotaxis and scototaxis are independent and specific to the individual. Therefore, these tests cannot replace each other, and a combination of behavioral tasks is recommended to best evaluate the effects of a given treatment on anxiety-like behavior.

## Figures and Tables

**Figure 1 animals-13-02481-f001:**
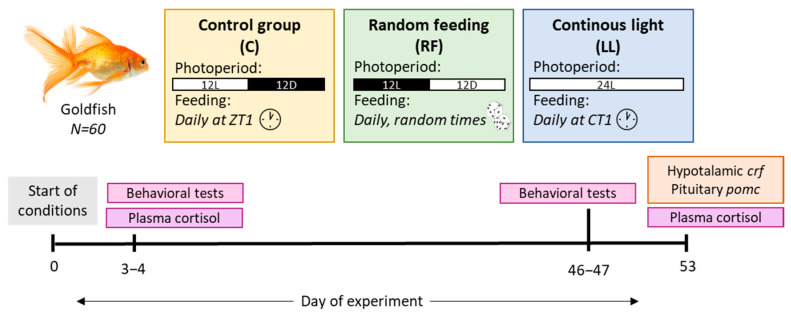
Diagram of experimental design 2 (for more information see text Section 2.2.2).

**Figure 2 animals-13-02481-f002:**
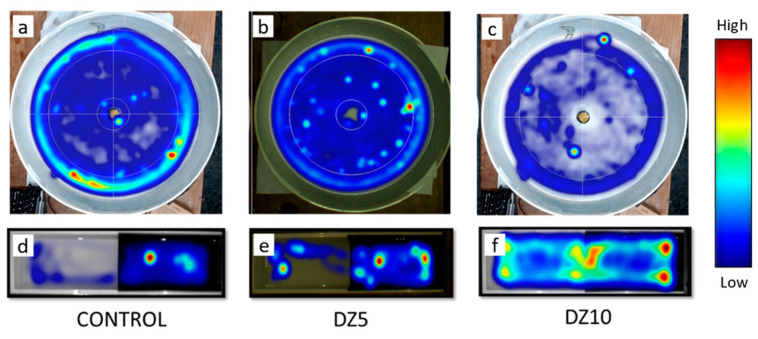
Heatmaps of the preferred zones in the open field test (**a**–**c**) and the black/white preference test (**d**–**f**) for control (**a**,**d**), diazepam 5 μg/g bw (DZ5, **b**,**e**) and diazepam 10 μg/g (DZ10, **c**,**f**) groups. Heatmaps indicate the areas of the arena where fish stayed for longer (mean of *n* = 8–16 fish/group) in a color scale of blue to red indicating low to high presence, respectively.

**Figure 3 animals-13-02481-f003:**
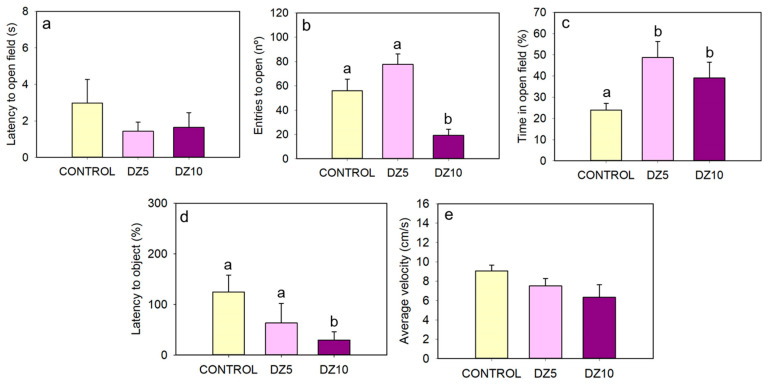
Effect of diazepam (DZ, 5 μg/g bw and 10 μg/g bw) in the open field with object approach test. (**a**) Latency to open field. (**b**) Entries to open. (**c**) Time in open field. (**d**) Latency to object. (**e**) Average velocity. Data are represented as mean + SEM (*n* = 8/group). Different letters indicate significant differences among groups (one-way ANOVA followed by Student–Newman–Keuls test).

**Figure 4 animals-13-02481-f004:**
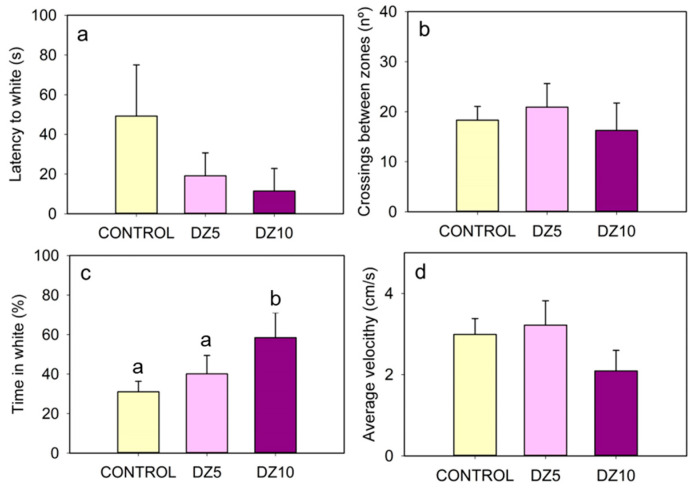
Effect of diazepam (DZ, 5 μg/g bw and 10 μg/g bw) in the black/white preference test. (**a**) Latency to white, (**b**) Crossings between zones, (**c**) Time in white, and (**d**) Average velocity. Data are represented as mean + SEM (*n* = 8/group). Different letters indicate significant differences among groups (one-way ANOVA followed by Student–Newman–Keuls test).

**Figure 5 animals-13-02481-f005:**
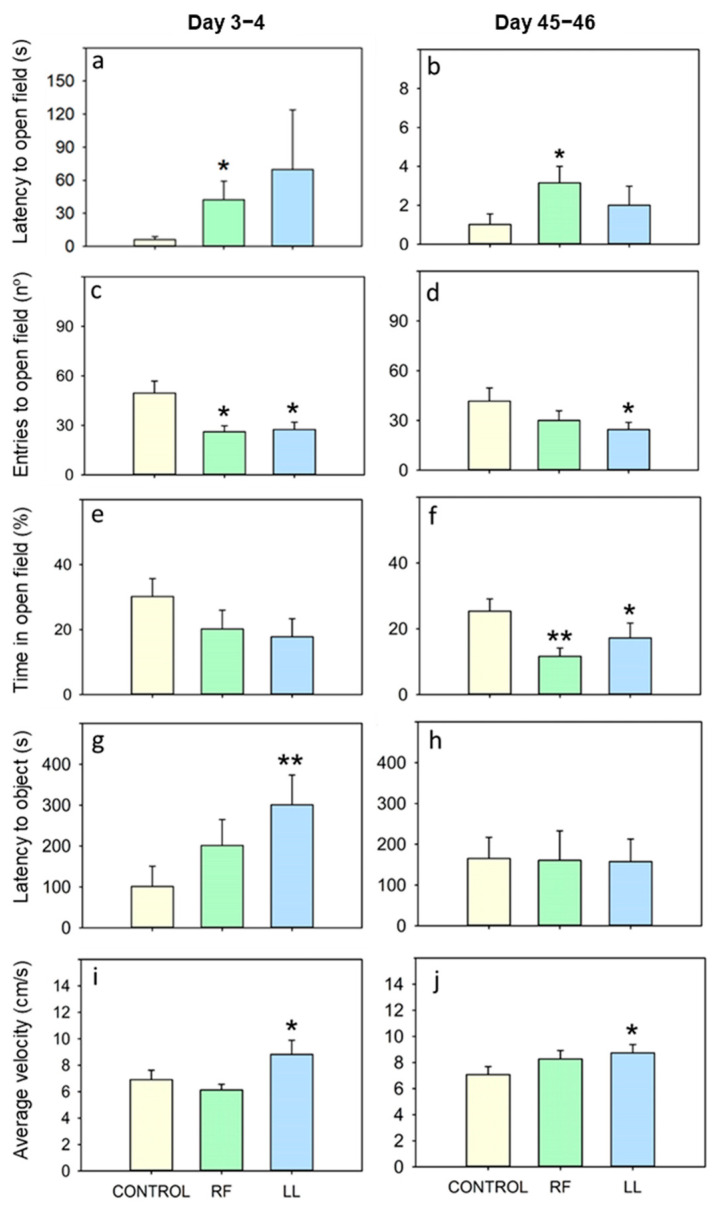
Effect of acute and chronic exposure to random feeding (RF) and continuous light (LL) in the open field with object approach test. (**a**,**b**) Latency to open field, (**c**,**d**) Entries to open field, (**e**,**f**) Time in open field, (**g**,**h**) Latency to object and (**i**,**j**) Average velocity. Data are represented as mean + SEM (*n* = 16/group), * *p* < 0.05, ** *p* < 0.01 (Student’s *t*-test or Mann–Whitney U test compared to control group).

**Figure 6 animals-13-02481-f006:**
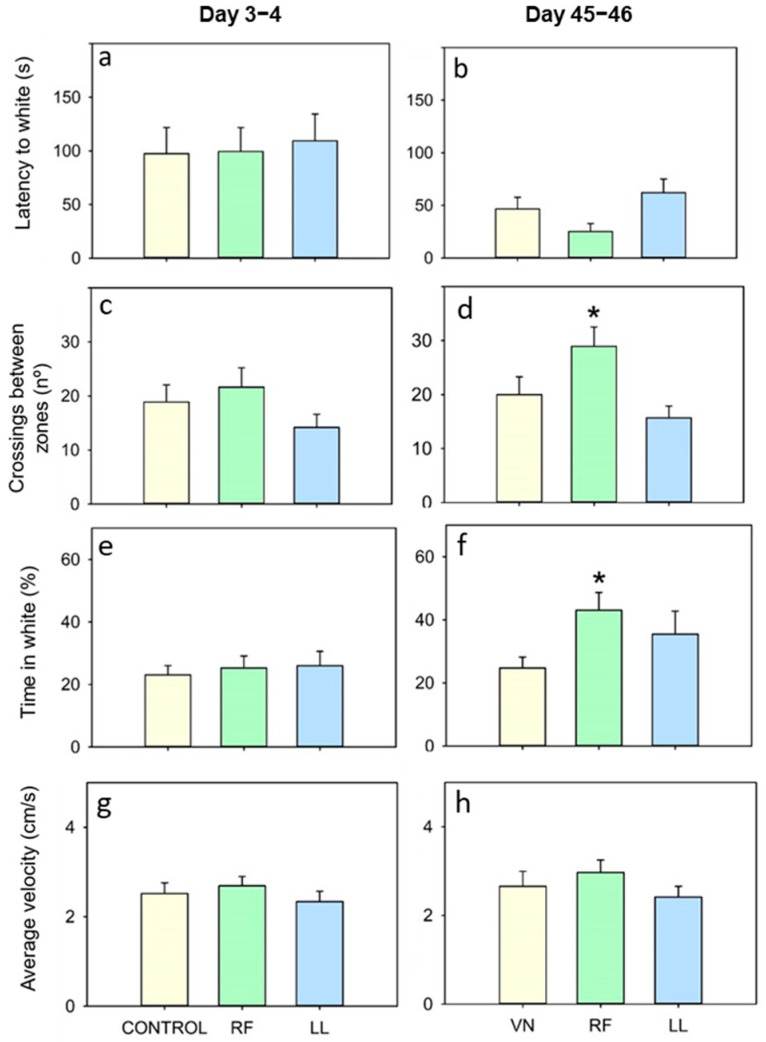
Effect of acute and chronic exposure to random feeding (RF) and continuous light (LL) in the black/white preference test. (**a**,**b**) Latency to white, (**c**,**d**) Crossings between zones, (**e**,**f**) Time in white, and (**g**,**h**) Average velocity. Data are represented as mean + SEM (*n* = 16/group). * *p* < 0.05 (Student’s *t*-test or Mann–Whitney U test compared to control group).

**Figure 7 animals-13-02481-f007:**
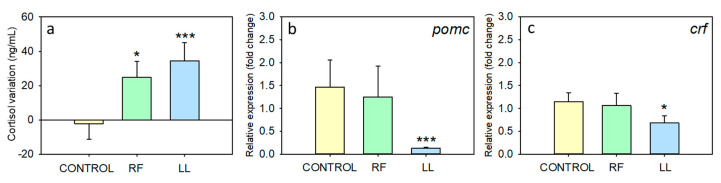
Parameters of hypothalamus–hypophysis–interrenal (HPI) axis activation in goldfish after chronic exposure to continuous light (LL) and random feeding (RF). (**a**) Individual variations between plasma cortisol levels on days 3–4 vs. day 53. (**b**) *pomc* expression in pituitary of goldfish on day 53. (**c**) *crf* expression in hypothalamus on day 53. Data are represented as mean + SEM (*n* = 16/group). * *p* < 0.05; *** *p* < 0.001 (Student’s *t*-test compared to control group).

**Table 1 animals-13-02481-t001:** Spearman correlations between behavioral parameters of the OF test and plasma cortisol levels in the same fish on days 3–4.

	Latency to Open Field	Entries to Open Field	Time in Open Field	Latency to Object	Average Velocity
Plasma cortisol	*p* = 0.63 CC = 0.07	*p* = 0.56 CC = 0.086	*p* = 0.56 CC = −0.08	*p* = 0.73 CC = 0.06	*p* = 0.76 CC = 0.04

(*n* = 48). CC = correlation coefficient.

**Table 2 animals-13-02481-t002:** Spearman correlations between behavioral parameters of the BW test and plasma cortisol levels in the same fish on days 3–4.

	Latency to White	Crossings between Zones	Time in White	Average Velocity
Plasma cortisol	*p* = 0.60 CC = 0.08	*p* = 0.39 CC = 0.13	*p* = 0.43 CC = −0.12	*p* = 0.76 CC = 0.04

(*n* = 48). CC= correlation coefficient.

**Table 3 animals-13-02481-t003:** Spearman correlations between behavioral parameters of the OF test in the same fish on days 3–4 and days 45–46.

	Latency to Open Field	Entries to Open Field	Time in Open Field	Latency to Object	Average Velocity
Days 3–4 vs. 45–46	*p* = 0.63 CC = −0.07	*p* = 0.04 CC = 0.31	*p* = 0.24 CC = 0.18	*p* = 0.03 CC = 0.37	*p* = 0.002 CC = 0.47

*p* < 0.05 (green) indicates significant relationship between the two variables (*n* = 48). CC = correlation coefficient.

**Table 4 animals-13-02481-t004:** Spearman correlations between behavioral parameters of the BW test performed by the same fish on days 3–4 and days 45–46.

	Latency to White	Crossings between Zones	Time in White	Average Velocity
Days 3–4 vs. 45–46	*p* = 0.04 CC = 0.32	*p* = 0. 35 CC = −0.15	*p* = 0.16 CC = 0.34	*p* = 0.06 CC = 0.29

*p* < 0.05 (green) indicates significant relationship between the two variables (*n* = 48). CC= correlation coefficient. Yellow indicates variables very close to statistical significance.

## Data Availability

Data are contained within this article or Appendix A.

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
