# Peer review of "Assessing Chronodisruption Distress in Goldfish: The Importance of Multimodal Approaches"

_animals, 2023, doi:10.3390/ani13152481_

Round 1

Reviewer 1 Report

The work described in this manuscript falls within the scope of animals and the experimental design is logically arranged. As a whole, the background of the study is well described and the methods used are novel. The highlights contain a summary of the manuscript's contents, giving a thorough point of view of what was studied as well as the methodology.

 1. Line 263, “ameaning”, incorrect writing.

2. Line135, how was the amount of feed adjusted over time to ensure that the daily ration remained at the same 1.5% BW as time progressed and the fish increased in size? Please add operational details.

3. Line148, what was the injected dose and proportions based on in this article?

4. How was the survival rate during the feeding period? If some of the fish in feeding treatment died during the trial, and mortality varied for the different treatments. This makes robust and reliable calculation of growth metrics difficult.

5. Line159, please unify the writing style of “5-L” or “5-liter” in the method section.

6. Line216, please unify the writing style of “min” or “minute” in the method section.

7. Line226, what amount does “a 10x volume of methanol was added” refers to?

8. A considerable amount of the article mentions circadian rhythm, but the calculation of some parameters does not involve rhythm related analysis, only correlation checking. These data support is insufficient. It is recommended to supplement the content of circadian rhythm analysis in terms of mRNA expression levels and other parameters as the Zeitgebers time changed, such as period, phase, and so on.

no

Reviewer 2 Report

The article is cogent and well written, all conclusions are well supported by the results, and I think it'll be of interest to a wide audience of readers. The only caveat is that I'd suggest presenting the data in the figures with boxplots rather than bars of means and SEMs, and as for statistics, I think ANOVA and post-hoc tests are a more correct approach than Student-t because there are more than two groups.

Round 2

Reviewer 1 Report

Revised according to review comments and can be accepted